# Double-Tongue Worm Shell Structure on Plastic Centrifugal Pump Performance Study

**Zhi Rao** *, **Lingfeng Tang and Hui Zhang**

School of Mechanical Engineering, Anhui Polytechnic University, Wuhu 241000, China
* Correspondence: 2220110145@stu.ahpu.edu.cn

**Abstract:** Aiming at the problem of high vibration and high wear of centrifugal pump tongue, this paper proposes a double-tongue volute structure. Under the condition of ensuring the reliability of CFD results, the influence of various combinations of tongue and volute base circle on the turbulent kinetic energy of centrifugal pump and the radial force of impeller is explored. The traditional single-tongue volute centrifugal pump is compared with various characteristic indexes, and the unsteady numerical calculation is carried out based on different working conditions. It is concluded that the double-tongue volute structure can improve the pressure fluctuation at the monitoring points near the tongue. The results show that the double-tongue volute structure can improve the static pressure gradient and velocity gradient of the middle section of the centrifugal pump and reduce the maximum turbulent kinetic energy value at the tongue under small flow conditions. When the working condition is 1.0 Q, the radial force of the impeller of the C-type double-tongue volute reaches the minimum value of 3.03 N, which can effectively balance part of the radial force.

**Keywords:** plastic centrifugal pump; double-tongue volute; steady calculation analysis; pressure pulsation

## 1. Declaration

UG (Unigraphics NX) is a product engineering solution produced by Siemens PLM Software Company, which provides users with digital modeling and verification methods for product design and processing. Unigraphics NX provides a proven solution for the user's virtual product design and process design needs and also meets various industrial needs. It is an interactive computer-aided design and computer-aided manufacturing system (CAD/CAM), it is powerful, and it can easily realize the construction of various complex entities and modeling.

ICEM is a professional CAE pre-processing software. It has powerful CAD model repair capabilities, automatic midplane extraction, unique grid 'sculpture' technology, grid editing technology, and extensive solver support capabilities.

FLUENT is a general purpose CFD software package, which is used to simulate the complex flow from incompressible to highly compressible. The best standard grid software with FLUENT is ICEM. FLUENT software contains a very rich and engineering-confirmed physical model. Due to the use of a variety of solution methods and multi-grid acceleration convergence technology, FLUENT can achieve the best convergence speed and solution accuracy. The flexible unstructured grid and solution-based adaptive grid technology and mature physical models can simulate hypersonic flow field, heat transfer and phase change, chemical reaction and combustion, multiphase flow, rotating machinery, dynamic/deformation grid, noise, material processing, and other complex mechanism of flow problems.

## 2. Introduction

The impeller and the worm casing, as the two essential components of the centrifugal pump, can cause the uneven flow of the internal fluid due to the dynamic coupling between

them, which may intensify the pressure pulsation in the pump, and this is often one of the factors that induce noise and vibration in the centrifugal pump; to alleviate this situation, it is urgent to investigate the spiral case structure of the centrifugal pump in depth [1].

Changliang Ye et al. [2] targeted a centrifugal pump with a guide vane and conducted numerical simulations of the flow field of a centrifugal pump under stall conditions and found that when the guide vane outlet is aligned with the centrifugal pump worm housing tongue, the corresponding head is the highest and the pressure fluctuation is the smallest. Shim Hyeon Seok et al. [3] investigated the effect of the number of vanes on the impeller–worm gear interaction and flow instability of a centrifugal pump. The results showed that the number of vanes affects performance parameters and local pressure fluctuations differently. Idris Muhammad Nuru [4] designed impellers using corrosive and non-corrosive materials to improve the reliability and efficiency of centrifugal pump impellers. Shamsuddeen Mohamed Murshid et al. [5] studied a double-suction double worm gear five-stage centrifugal pump with different baffles installed at the double worm casing. The comparison revealed that installing baffles enhanced the head and efficiency of the pump. Zemanová L. et al. [6] built an experimental rig including an actual impeller to observe the flow in the back sidewall gap by optical methods and to evaluate the axial thrust and torque, introducing a new flow pattern diagram. Jaiswal Ashutosh Kumar et al. [7] used a multi-objective genetic algorithm to find the optimal design point to optimize the input power of a centrifugal pump by varying the impeller vane exit angle. Uralov Bakhtiyor et al. [8] studied the effect of impeller blade hydraulic abrasive wear on the centrifugal pump head and the wear of centrifugal pump components under natural conditions and gave the dependence of wear on its characteristic dimensions and operating duration. Klimentov Kliment et al. [9] validated CFD models of two pump impellers and performed a numerical simulation study of the variation of the slip coefficient during the dressing process, obtaining equations describing the relationship between the slip coefficient and the dressing rate of the studied pumps. In this paper, the design of a single-tongue worm casing structure was changed to a double-tongue worm casing structure by changing the tongue type, and the spacing value between the worm casing base circle and the tongue was used as the study object to investigate the effect of different spacing values on the performance of plastic centrifugal pumps.

This paper takes a plastic centrifugal pump as the research object, creates the centrifugal pump computational domain model by using UG, completes the mesh division and irrelevance verification based on ICEM, determines the mesh quantity of the centrifugal pump models with five worm casing structures, and uses FLUENT to calculate the five worm casing pump models with constant, analyzes and compares the difference between single-tongue volute centrifugal pump and double-tongue volute centrifugal pump from four perspectives, such as static pressure, velocity, turbulent kinetic energy, and impeller radial force under different working conditions. The difference between the centrifugal pump with single-tongue volute and the centrifugal pump with double-tongue volute is analyzed and compared, and the difference between the centrifugal pumps with different pitch values of the volute base circle and the tongue is analyzed. It is concluded that the double-tongue volute structure can improve the static pressure gradient distribution and velocity gradient distribution of the middle section of the centrifugal pump, reduce the turbulent kinetic energy value at the tongue, and balance part of the radial force. Using FLUENT for unsteady numerical calculation, it is concluded that the double-tongue volute structure can improve the pressure pulsation in the centrifugal pump, and the influence of the double-tongue volute structure on the performance of the plastic centrifugal pump is completed.

### 3. Structure Design of Double-Tongue Volute Plastic Centrifugal Pump

In plastic centrifugal pumps, the small gap between the tongue and the vane outlet leads to complex flow in this region, affecting the performance of the centrifugal pump. This section takes the tongue type as the object of study and adjustw the spacing between the tongue and the base circle of the worm casing, evaluating five types of centrifugal pump structures and completing the design of the plastic centrifugal pump-related systems.

#### 3.1. Impeller Structure Design

The impeller is one of the vital over-flow components of the centrifugal pump, and the velocity coefficient method [10] is applied to its structural design. Its main parameters are shown in Table 1.

**Table 1.** Impeller main parameters.

| Geometric Parameters | Numerical Value | Geometric Parameters | Numerical Value |
|---|---|---|---|
| Impeller inlet diameter $D_1$ | 80 mm | Impeller inlet placement angle, $\beta_1$ | 25° |
| Blade inlet diameter $D_2$ | 258 mm | Blade exit placement angle, $\beta_2$ | 30° |
| Blade wrap angle $\varphi$ | 120° | Number of blades, Z | 6 |

The drawing of the blade profile directly affects the fluid flow in the impeller runner, and the trajectory of the liquid in the impeller runner is spiral, so the blade profile is drawn based on the equal-variable-angle logarithmic spiral method, and its drawing principle is shown in Figure 1.

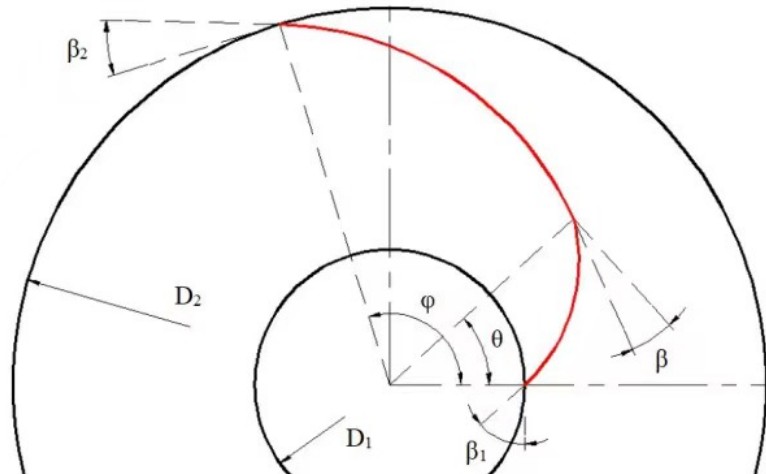

**Figure 1.** Equal-variable-angle logarithmic helix method blade profile drawing principle diagram.

The equation of the blade profile of the impeller is shown in Equation (1):

$$r = \frac{1}{2}D_1 \left( \frac{\cos \beta_1}{\cos \beta} \right)^{\frac{\varphi}{\beta_2 - \beta_1}} \tag{1}$$

The blade profile can be obtained by substituting $\beta_1 = 25°$, $\beta_2 = 30°$, $D_1 = 80$ mm, and $\varphi = 120°$ into Equation (1). In total, 50 fitting points are selected, and the blade profile obtained is plotted using MATLAB R2019a software, where the ordinate and abscissa represent the coordinates of the fitting points, as seen in Figure 2.

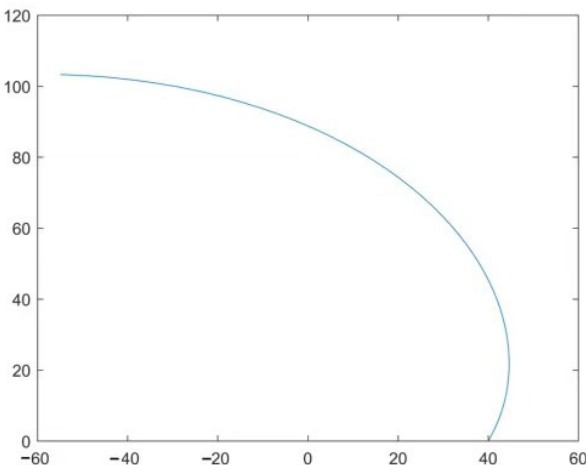

**Figure 2.** Blade profile diagram.

### 3.2. Volute Structure Design

The diameter $D_3$ of the volute base circle should be slightly larger than the diameter $D_2$ of the impeller outlet, but if the value is too large, it will affect the pump's efficiency. If the value is too small, it is easy to block the fluid and cause the pump's vibration and noise. The volute inlet width $b_3$ ensures a particular gap between the impeller and the volute. After calculation, the volute base circle diameter is 270 mm, the volute inlet width is 50 mm, and the tongue placement angle is 15°. The area of section VIII of the volute is 8.772 cm². UG is used to create a three-dimensional model of the calculation domain of each part of the centrifugal pump, as shown in Figures 3 and 4. The geometric parameters of types B, C, D, and E are shown in Tables 2–5.

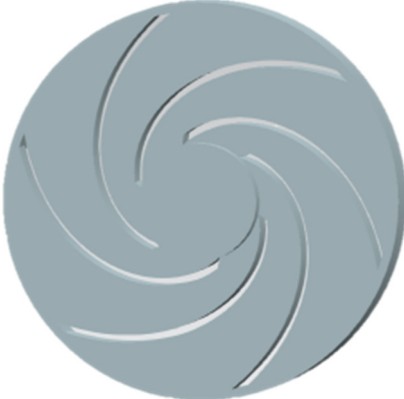

**Figure 3.** Impeller fluid domain model diagram.

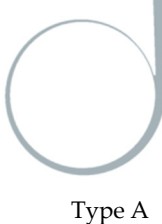
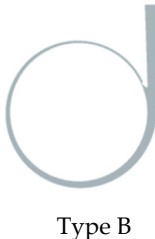
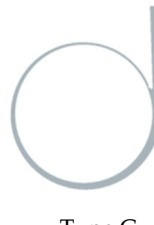
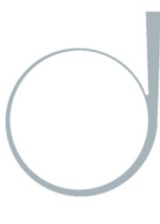
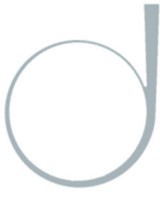

| Type A | Type B | Type C | Type D | Type E |

**Figure 4.** Model diagram of five types of worm shell structures.

**Table 2.** Table of main geometric parameters of B-type volute.

| Geometric Parameter | Numerical Value | Geometric Parameter | Numerical Value |
|---|---|---|---|
| $D_3$ | 270 mm | $b_3$ | 50 mm |
| $\delta$ | 1 mm | $R_1$ | 3.4 mm |
| $R_2$ | 2.8 mm | $\varphi_1$ | 20° |
| $x$ | 127.8 mm | $y$ | 55.7 mm |

**Table 3.** Table of main geometric parameters of C-type volute.

| Geometric Parameter | Numerical Value | Geometric Parameter | Numerical Value |
|---|---|---|---|
| $D_3$ | 270 mm | $b_3$ | 50 mm |
| $\delta$ | 2 mm | $R_1$ | 2.4 mm |
| $R_2$ | 2.8 mm | $\varphi_1$ | 20° |
| $x$ | 128.7 mm | $y$ | 53.5 mm |

**Table 4.** Table of main geometric parameters of D-type volute.

| Geometric Parameter | Numerical Value | Geometric Parameter | Numerical Value |
|---|---|---|---|
| $D_3$ | 270 mm | $b_3$ | 50 mm |
| $\delta$ | 3 mm | $R_1$ | 2.8 mm |
| $R_2$ | 2.8 mm | $\varphi_1$ | 20° |
| $x$ | 129.7 mm | $y$ | 54.0 mm |

**Table 5.** Table of main geometric parameters of E-type volute.

| Geometric Parameter | Numerical Value | Geometric Parameter | Numerical Value |
|---|---|---|---|
| $D_3$ | 270 mm | $b_3$ | 50 mm |
| $\delta$ | 4 mm | $R_1$ | 2.8 mm |
| $R_2$ | 2.8 mm | $\varphi_1$ | 20° |
| $x$ | 130.6 mm | $y$ | 54.5 mm |

The assembled plastic centrifugal pump fluid calculation domain model is shown in Figure 5.

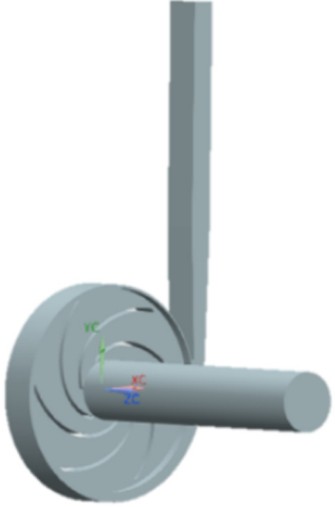

**Figure 5.** Plastic centrifugal pump fluid calculation domain model diagram.

The plastic centrifugal pump fluid computational domain model is unstructured meshed with the help of ICEM, and the final mesh is shown in Figure 6. Each part of the plastic centrifugal pump fluid computational domain model meshes separately, and the final mesh file is combined.

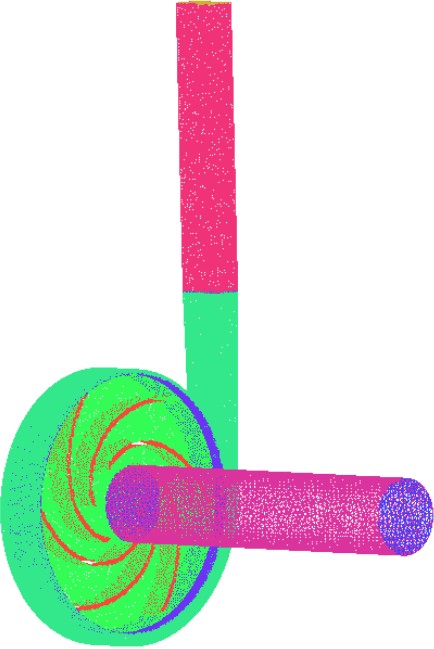

**Figure 6.** Plastic centrifugal pump fluid calculation domain model grid map.

Taking the A-type worm shell fluid computational domain model as an example, six groups of different numbers of meshes are set for numerical computation to complete the mesh-independent verification work, and the results are shown in Table 6.

**Table 6.** Table of mesh-independent verification results of A-type worm shell.

| Programs | Number of Cells | Lift |
|---|---|---|
| 1 | 964,606 | 24.59 m |
| 2 | 1,009,473 | 24.41 m |
| 3 | 1,065,227 | 24.25 m |
| 4 | 1,132,557 | 24.15 m |
| 5 | 1,184,111 | 24.13 m |
| 6 | 1,254,375 | 24.12 m |

Table 6 and Figure 7 show that when the grid quantity of the centrifugal pump model with an A-type worm shell is about 1.2 million, the influence on the centrifugal pump head value is small. When the grid quantity exceeds 1.2 million, the impact on the centrifugal pump head value is tiny. Taking into account, the division method with the total number of cells of 1,184,111 is chosen to mesh the calculation domain of other pump models, and the final determination of the number of cells for the five worm shell structure centrifugal pump models is shown in Table 7. In the next chapter, we will introduce the double-tongue plastic centrifugal pump constant calculation analysis.

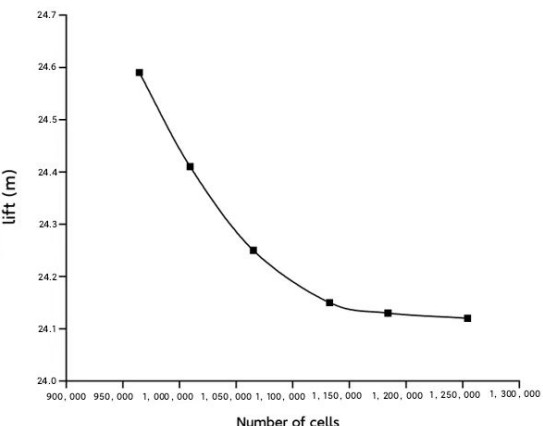

**Figure 7.** Mesh-independent verification diagram of A-type worm shell.

**Table 7.** Grid statistics of five worm shell structure centrifugal pump models.

| Snail Shell Type | Number of Cells |
| --- | --- |
| Type A | 1,184,111 |
| Type B | 1,195,154 |
| Type C | 1,195,560 |
| Type D | 1,195,629 |
| Type E | 1,195,783 |

## 4. Double-Tongue Plastic Centrifugal Pump Constant Calculation Analysis

To investigate the effect of the spacing between the worm casing base circle and the tongue of the plastic centrifugal pump on the performance of the plastic centrifugal pump, with the help of FLUENT, the steady calculation of the pump model of each volute structure is carried out. The differences between the single-tongue and double-tongue centrifugal pumps and between centrifugal pumps with different spacing values between the worm casing base circle and tongue are analyzed and compared from four perspectives: static pressure, velocity, turbulent kinetic energy, and impeller radial force under different operating conditions.

### 4.1. Static Pressure Distribution of Different Worm Shell Structures under Various Operating Conditions

1.　The static pressure distribution in the cross-section of the pump under 0.8 Q conditions

The static pressure distribution of the cross-section in the pump at 0.8 Q operating conditions is shown in Figure 8.

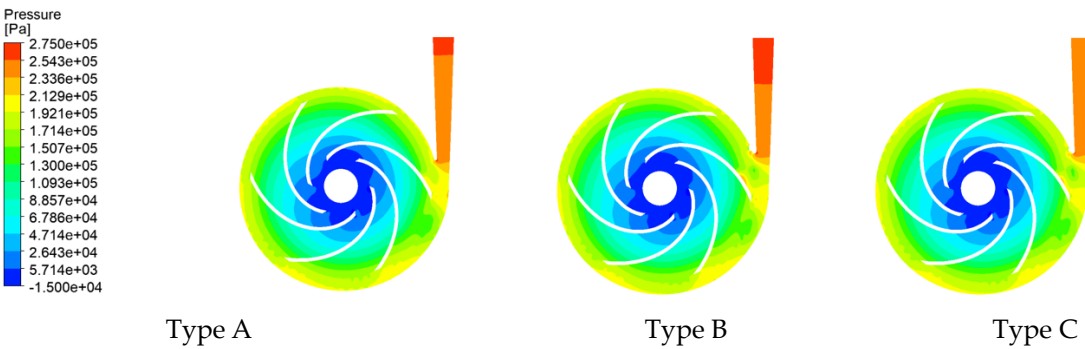

Type A　　　　　　　　　　　Type B　　　　　　　　　　Type C

**Figure 8.** *Cont*.

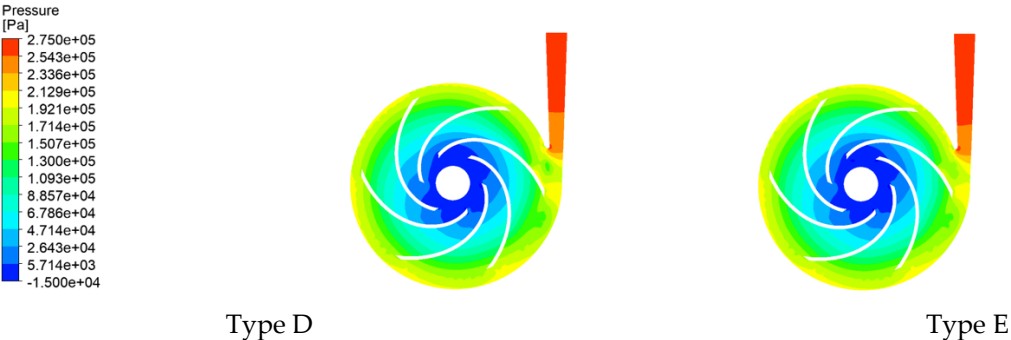

**Figure 8.** Static pressure distribution in the pump section under 0.8 Q working conditions.

Under this condition, the static pressure distribution in the impeller passage of the single-tongue volute centrifugal pump and the double-tongue volute centrifugal pump is similar [11]. For each model pump, the minimum values of static pressure are found at the inlet of the impeller, and the minimum values of static pressure are −29,787.6 Pa, −30,825.4 Pa, −30,866.8 Pa, −31,887.8 Pa, and −31,017.2 Pa. As the impeller rotates continuously and the fluid gradually enters the interior of the pump, the static pressure values increase progressively and reach larger values at the impeller outlet.

There are differences in the static pressure distribution in the volute flow channel of the single-tongue volute centrifugal pump and the double-tongue volute centrifugal pump. The stationary pressure distribution between different sections of the volute is relatively uniform. Still, the static pressure value at each volute section is slightly larger than the static pressure value between each section. In contrast, the stationary pressure gradient distribution is significant between section VIII of the volute and the outlet of the diffusion section. The static pressure value reaches the maximum at the outlet. The stationary pressure distribution of the volute diffusion section of different structural types is quite different. The static pressure values at the outlet are 277,546 Pa, 270,260 Pa, 261,837 Pa, 272,084 Pa, and 278,230 Pa, respectively.

By comparing the models, it can be found that the static pressure gradient distribution of the C-type double-tongue volute centrifugal pump in the diffusion section is improved compared with the A-type single-tongue volute centrifugal pump. The static pressure gradient phenomenon of the other three types of double-tongue volute centrifugal pumps B, D, and E is more evident than that of the A-type single-tongue volute centrifugal pump in the diffusion section, and the static pressure value at the outlet increases. However, the static pressure value decreases from the VIII section of the volute to the volute tongue, and the stationary pressure distribution here is partially improved.

2. The static pressure distribution of the cross-section in the pump under 1.0 Q conditions

The static pressure distribution of the cross-section in the pump under 1.0 Q operating conditions is shown in Figure 9.

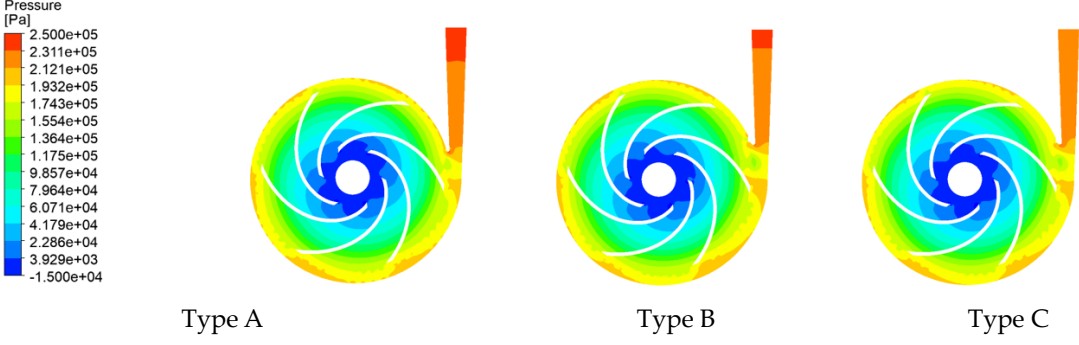

**Figure 9.** *Cont.*

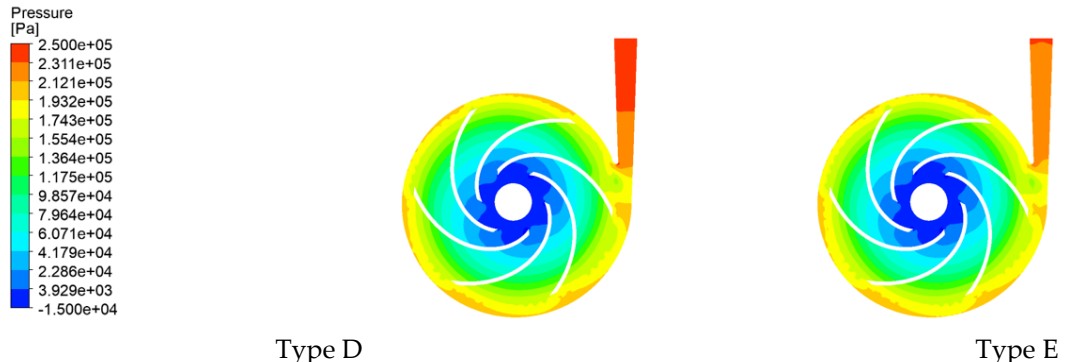

**Figure 9.** Static pressure distribution in the pump section under 1.0 Q working conditions.

Under this condition, the static pressure distribution in the impeller passage of the single-tongue volute centrifugal pump and the double-tongue volute centrifugal pump is similar [12]. For each model pump, the minimum static pressure appears at the inlet of the impeller, and the minimum static pressure is −31,102.4 Pa, −31,160.0 Pa, −32,147.1 Pa, −34,339.8 Pa, and −30,345.9 Pa, respectively. As the rotating fluid of the impeller begins to enter the interior of the pump, the static pressure value gradually increases and reaches a more significant value at the outlet of the impeller.

There are differences in the static pressure distribution in the volute flow channel of the single-tongue volute centrifugal pump and the double-tongue volute centrifugal pump. The stationary pressure distribution between different sections of the volute is relatively uniform. From the tongue of the volute, the static pressure value near the volute wall shows a gradual increase trend, and the static pressure value of the outer wall of the volute shows a trend greater than the static pressure value of the inner wall. There is a significant stationary pressure gradient distribution between the VIII section of the volute and the outlet of the diffusion section, and the static pressure value reaches the maximum at the outlet. The stationary pressure distribution of the volute diffusion section of different structural types is quite different. The static pressure values at the outlet are 243,105 Pa, 235,444 Pa, 235,482 Pa, 247,938 Pa, and 234,203 Pa, respectively.

Compared with each model, it can be found that the static pressure gradient distribution of B-, C-, and E-type double-tongue volute centrifugal pumps is improved compared with that of A-type single-tongue volute centrifugal pumps. Compared with the A-type single-tongue volute, the static pressure gradient of the D-type double-tongue volute centrifugal pump is more evident in the diffusion section, and the static pressure value at the outlet increases. However, the static pressure value decreases from the VIII section of the volute to the volute tongue, and the stationary pressure distribution is partially improved.

3.    The static pressure distribution of the cross-section in the pump under 1.2 Q conditions

The static pressure distribution of the cross-section in the pump under 1.2 Q working conditions is shown in Figure 10.

Under this condition, the static pressure distribution in the impeller passage of the single-tongue volute centrifugal pump and the double-tongue volute centrifugal pump is similar [13]. For each model pump, the minimum static pressure appears at the inlet of the impeller, and the minimum static pressure is −33,210.3 Pa, −32,642 Pa, −32,911.5 Pa, −32,556.3 Pa, and −31,447.1 Pa, respectively. With the continuous rotation of the impeller, the static pressure value gradually increases and reaches a more significant value at the outlet of the impeller.

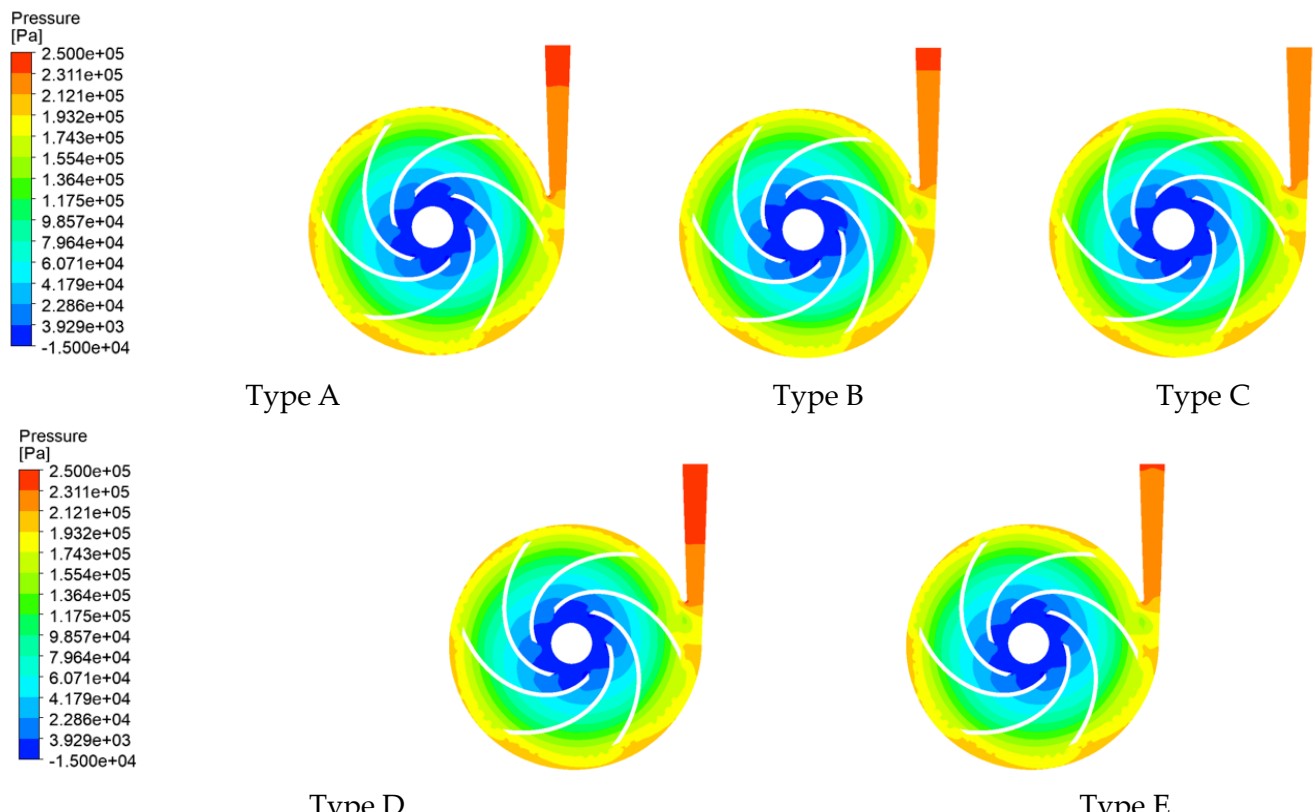

**Figure 10.** Static pressure distribution in the pump section under 1.2 Q working conditions.

There are differences in the static pressure distribution in the volute flow channel of the single-tongue volute centrifugal pump and the double-tongue volute centrifugal pump. The stationary pressure distribution between different sections of the volute is relatively uniform. From the tongue of the volute, the static pressure value near the volute wall shows a gradual increase trend, and the pressure value of the outer wall of the volute shows a trend greater than the static pressure value of the inner wall. There is a significant stationary pressure gradient distribution between the VIII section of the volute and the outlet of the diffusion section, and the static pressure value reaches the maximum at the outlet. The stationary pressure distribution of the volute diffusion section of different structural types is quite different. The static pressure values at the outlet are 223,178 Pa, 216,211 Pa, 217,546 Pa, 217,588 Pa, and 217,486 Pa, respectively.

Compared with each model, it can be found that the static pressure gradient distribution of B, C, and E double-tongue volute centrifugal pumps is improved compared with that of A single-tongue volute in the diffusion section. Compared with the A-type single-tongue volute centrifugal pump, the static pressure gradient phenomenon of the D-type double-tongue volute centrifugal pump is more severe in the diffusion section, and the static pressure value at the outlet increases. However, the static pressure value decreases from the VIII section of the volute to the volute tongue, and the stationary pressure distribution is partially improved.

### 4.2. Sectional Velocity Distribution of Different Volute Structures under Various Working Conditions

1.  The velocity distribution of the cross-section in the pump under 0.8 Q conditions

    The velocity distribution of the cross-section in the pump at 0.8 Q operating conditions is shown in Figure 11.

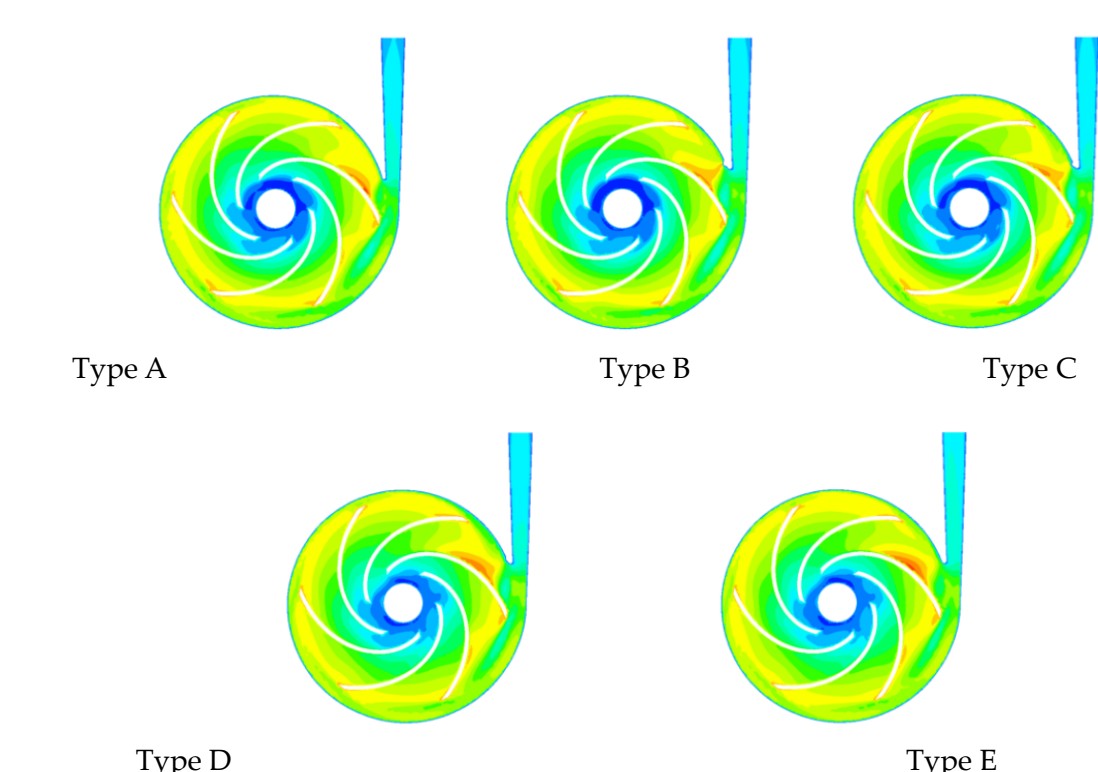

**Figure 11.** Velocity distribution of cross-section in the pump under 0.8 Q working conditions.

Under this condition, the middle section velocity distribution of the single-tongue volute centrifugal pump and the double-tongue volute centrifugal pump is roughly the same. The impeller passage has a local low-speed region with a small range of zero velocity near the impeller inlet. As the impeller rotates, the internal fluid velocity increases along the radial direction of the impeller. In the volute channel, there is a significant velocity gradient between the VIII section and the outlet of the diffusion section. There is also a small range of low-speed areas at the outlet of some volute structures. In addition, among the five structural types of volutes, there is a small range of high-speed sites on the back of the blade near the tongue, and the maximum speed appears here, reaching 22.57 m/s, 20.71 m/s, 20.63 m/s, 20.63 m/s, and 21.38 m/s, respectively. There is a small area of low speed between the VII and VIII sections of the volute.

It can be found from the comparison model that in the diffusion section, the low-speed zone of the centrifugal pump with double-tongue volute is smaller than that of the centrifugal pump with single-tongue volute, and the low-speed area near the tongue increases slightly.

2.  The velocity distribution of the cross-section in the pump under 1.0 Q conditions

    The velocity distribution of the cross-section in the pump at 1.0 Q operating conditions is shown in Figure 12.

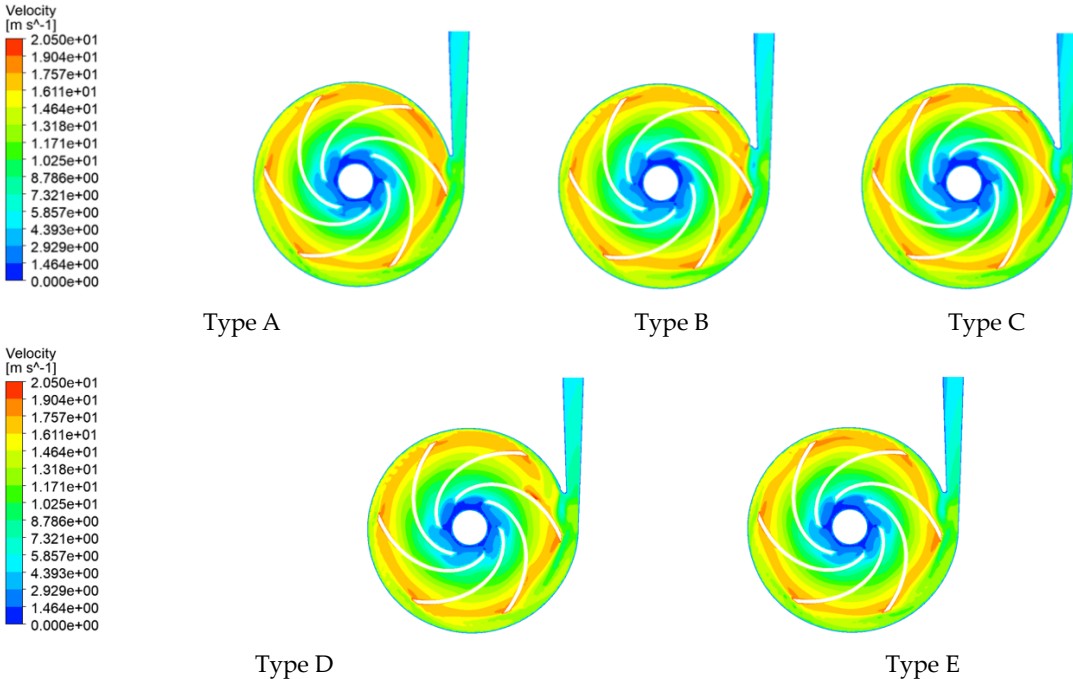

**Figure 12.** Velocity distribution of the cross-section in the pump under 1.0 Q working conditions.

Under this condition, the middle section velocity distribution of the single-tongue volute centrifugal pump and the double-tongue volute centrifugal pump is roughly the same. The impeller passage has a local low-speed region with a small range of zero velocity near the impeller inlet. As the impeller rotates, the internal fluid velocity increases radially along the impeller and reaches a maximum at the junction of the impeller and the volute. In the volute channel, there is a significant velocity gradient between the VIII section and the outlet of the diffusion section. There is also a small range of low-speed areas at the outlet of some volute structures. In addition, there are different degrees of low-speed regions near the tongue of the five structural types of volutes, and the low-speed region in the B-type volute is larger.

Compared with each model, it can be found that the double-tongue volute centrifugal pump is significantly reduced in the high-speed area near the tongue compared with the single-tongue volute centrifugal pump, and the velocity distribution in the diffusion section is minimal.

3. The velocity distribution of cross-section in pump under 1.2 Q condition

The velocity distribution of the cross-section in the pump under 1.2 Q working conditions is shown in Figure 13.

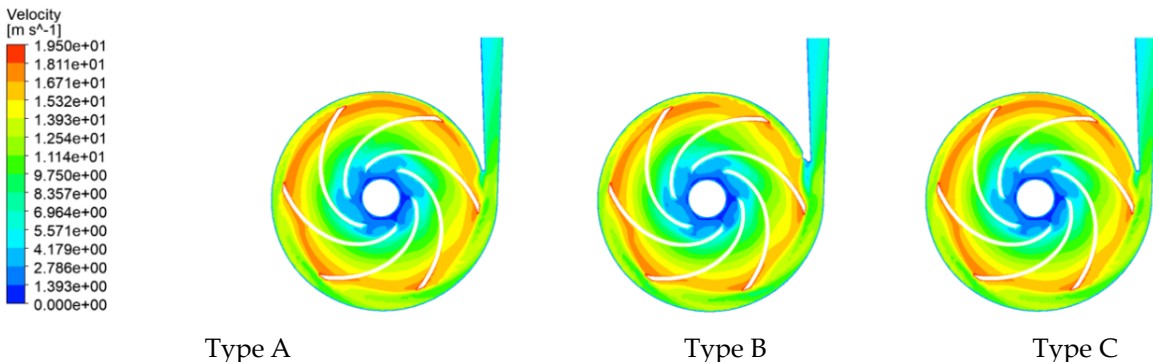

**Figure 13.** *Cont.*

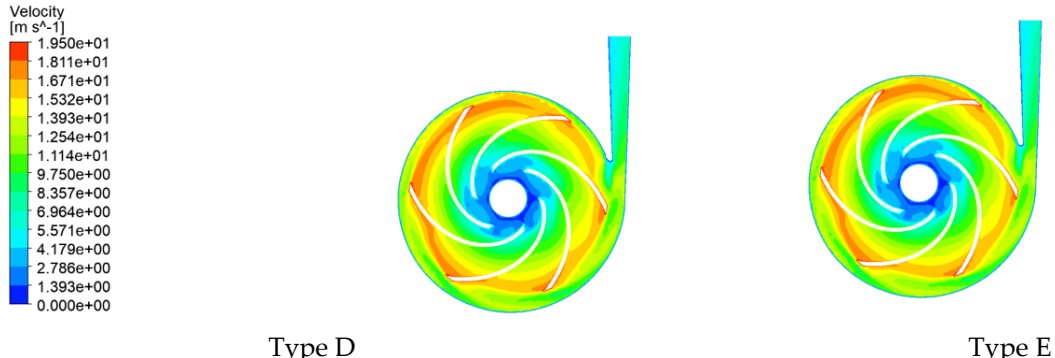

**Figure 13.** Velocity distribution of the cross-section in the pump under 1.2 Q working conditions.

Under this condition, the middle section velocity distribution of the single-tongue volute centrifugal pump and the double-tongue volute centrifugal pump is roughly the same. The impeller passage has a local low-speed region with a small range of zero velocity near the impeller inlet. As the impeller rotates, the internal fluid velocity increases radially along the impeller and reaches a maximum at the junction of the impeller and the volute. In the volute channel, there is a significant velocity gradient between the VIII section and the outlet of the diffusion section, and the small-scale low-speed region at the outlet of some volute structures disappears. In addition, there are different degrees of low-speed regions near the tongue of the five structural types of volutes, and the low-speed region in the B-type volute is larger.

Compared with each model, it can be found that the high-speed area near the tongue of the double-tongue volute centrifugal pump is significantly smaller than that of the single-tongue volute centrifugal pump, and the velocity distribution in the diffusion section is minimal. It shows that the volute of the double-tongue volute structure has a more noticeable effect on improving the velocity near the tongue, which is more conducive to the flow of the fluid in the centrifugal pump.

*4.3. Turbulent Kinetic Energy Distribution of Different Worm Shell Structures under Each Working Condition*

1. The distribution of turbulent kinetic energy in the cross-section of the pump under 0.8 Q conditions

The turbulent kinetic energy distribution of the cross-section in the pump at 0.8 Q operating conditions is shown in Figure 14.

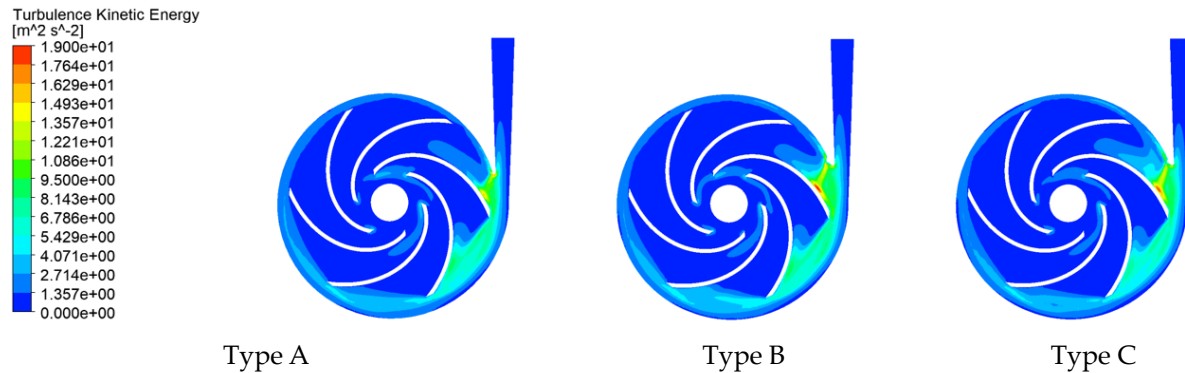

**Figure 14.** *Cont.*

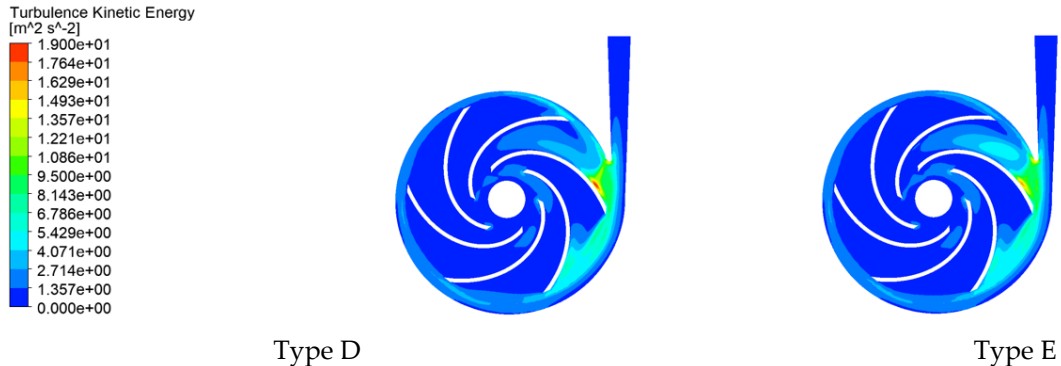

**Figure 14.** Turbulent kinetic energy distribution of the cross-section in the pump under 0.8 Q working conditions.

Under this working condition, the turbulent kinetic energy of the middle section of the single-tongue volute centrifugal pump and the double-tongue volute centrifugal pump is roughly the same. The turbulent kinetic energy at the inlet of the impeller is not much different, and the turbulent kinetic energy distribution in the other blade channels of the impeller is further. The turbulent kinetic energy distribution at the impeller outlet, volute sections, and volute diffusion section is quite different.

Comparing the centrifugal pumps with five types of volutes, it can be found that the turbulent kinetic energy of the double-tongue volute centrifugal pump is significantly higher than that of the single-tongue volute centrifugal pump near the volute tongue, and the turbulent kinetic energy value here reaches the maximum value, which indicates that the double-tongue has a more significant influence on the turbulent kinetic energy distribution near the volute tongue.

2. The distribution of turbulent kinetic energy in the cross-section of the pump under 1.0 Q conditions

The turbulent kinetic energy distribution of the cross-section in the pump at 1.0 Q operating conditions is shown in Figure 15.

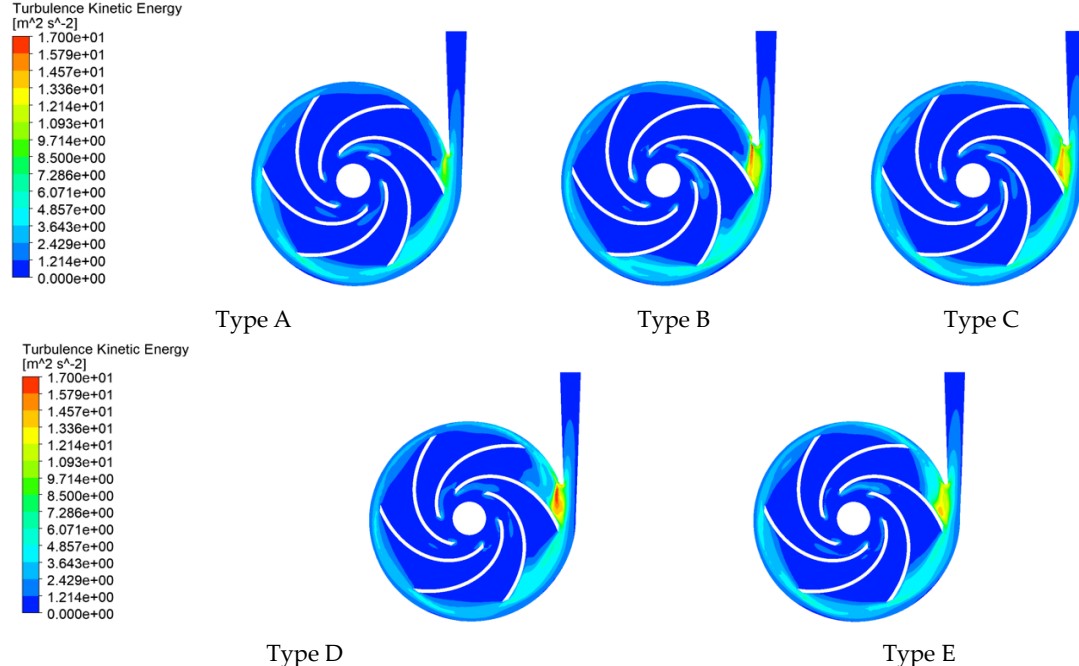

**Figure 15.** Turbulent kinetic energy distribution in the pump section under 1.0 Q working conditions.

Under this condition, the turbulent kinetic energy of the middle section of the single-tongue volute centrifugal pump and the double-tongue volute centrifugal pump is roughly the same. The turbulent kinetic energy at the inlet of the impeller is not much different. In contrast, the turbulent kinetic energy distribution in the other blade channels of the impeller is further. The turbulent kinetic energy distribution at the impeller outlet, volute sections, and volute diffusion section is quite different.

Comparing the centrifugal pumps with five types of volutes, it can be found that the turbulent kinetic energy of the centrifugal pump with double-tongue volute is significantly larger than that of the centrifugal pump with single-tongue volute near the volute tongue, and the turbulent kinetic energy value reaches the maximum value here, which indicates that the double-tongue has a significant influence on the turbulent kinetic energy distribution near the volute tongue. The D-type volute has the most critical influence. In the next chapter, we will introduce the unsteady calculation and analysis of double-tongue plastic centrifugal pump.

3. The distribution of turbulent kinetic energy in the cross-section of the pump under 1.2 Q conditions

The turbulent kinetic energy distribution of the cross-section in the pump at 1.2 Q operating conditions is shown in Figure 16.

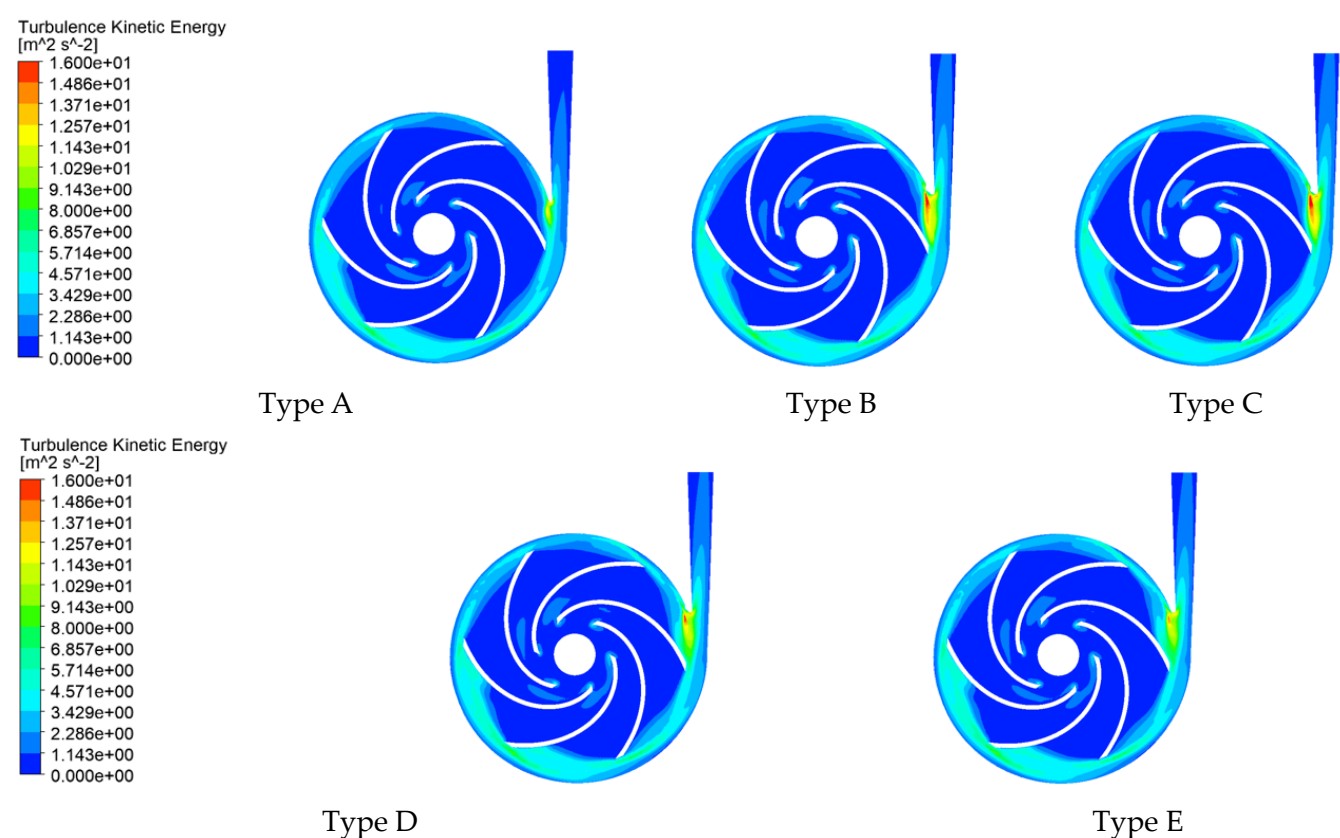

**Figure 16.** Turbulent kinetic energy distribution of the cross-section in the pump under 1.2 Q working conditions.

Under this working condition, the turbulent kinetic energy of the middle section of the single-tongue volute centrifugal pump and the double-tongue volute centrifugal pump is roughly the same. The turbulent kinetic energy at the inlet of the impeller is not much different, and the turbulent kinetic energy distribution in the different blade channels of the impeller is different. The turbulent kinetic energy distribution at the impeller outlet, volute sections, and volute diffusion section is quite different.

Comparing the centrifugal pumps with five types of volutes, it can be found that the turbulent kinetic energy of the double-tongue volute centrifugal pump is significantly higher than that of the single-tongue volute centrifugal pump near the volute tongue, and the turbulent kinetic energy value here reaches the maximum value, which indicates that the double-tongue volute has a more significant influence on the turbulent kinetic energy distribution near the volute tongue. The D and E volutes have less influence, and the B and C volutes have a more significant impact.

### 4.4. Analysis of Impeller Radial Force under Each Working Condition with Different Worm Shell Structures

The radial force of the impeller can be obtained by numerical integration of the pressure and viscous force on the circumference of the impeller, and the emphasis on the two components can be obtained by the Pythagorean theorem [14]. With the help of FLUENT, the radial force values on these two components can be directly obtained. The resultant force of the impeller radial force can be obtained after calculation, as shown in Table 8.

**Table 8.** Impeller radial force table.

| Work Conditions | A Type of Worm Shell | B-Type Worm Shell | C-Type Worm Shell | D-Type Worm Shell | E-Type Worm Shell |
|---|---|---|---|---|---|
| 0.8 Q | 57.03 N | 49.65 N | 51.35 N | 65.70 N | 63.78 N |
| 1.0 Q | 6.41 N | 3.92 N | 3.03 N | 8.67 N | 7.35 N |
| 1.2 Q | 20.15 N | 14.69 N | 17.26 N | 16.37 N | 18.64 N |

Table 8 shows that each volute centrifugal pump's impeller radial force value is the smallest at 1.0 Q working condition, which is less than 10 N. Still, theoretically, the impeller radial force value should be zero at this working condition [15]. The remaining deviations from the design condition have larger impeller radial force values, and the more departures from the design condition, the larger the impeller radial force value is [16].

Under the condition of 0.8 Q, the radial force of the impeller of B and C double-tongue volute centrifugal pump is slightly smaller than that of A single-tongue volute centrifugal pump, which is reduced by 13% and 10%, respectively. The radial force value of the impeller of the D and E double-tongue volute centrifugal pump is slightly larger than that of the A single-tongue volute centrifugal pump, which is increased by 18% and 12%, respectively.

Under the condition of 1.0 Q, the radial force of the impeller of B- and C-type double-tongue volute centrifugal pumps is slightly smaller than that of single-type tongue volute centrifugal pumps, which is reduced by 39% and 99%, respectively. The radial force value of the impeller of the D and E double-tongue volute centrifugal pump is slightly larger than that of the A single-tongue volute centrifugal pump, which is increased by 35% and 15%, respectively. Under this condition, the radial force of the impeller of the C-type double-tongue volute reaches a minimum value of 3.03 N.

Under the condition of 1.2 Q, the impeller radial force values of the B-, C-, D-, and E-type double-tongue volute centrifugal pumps are less than that of the A-type single-tongue volute centrifugal pump, which is reduced by 27%, 14%, 19%, and 7%, respectively.

Therefore, the double-tongue volute structure can reduce the radial force of the impeller to a certain extent and achieve the effect of balancing the radial force of the part.

### 4.5. External Characteristic Experiment of Plastic Centrifugal Pump

Combined with the results of steady calculation, among the four double-tongue volute structures, the internal flow field of the C-type double-tongue volute is better than that of the other three double-tongue volute structures. Therefore, the external characteristics of the A-type single-tongue volute and the C-type double-tongue volute were tested for the plastic centrifugal pump.

After the external characteristic experiment of the plastic centrifugal pump is completed, the relevant experimental data are derived from the 'pump product test system' software, and the comparison diagram of the external characteristic curve of the single- and double-tongue volute centrifugal pump shown in Figure 17 is obtained.

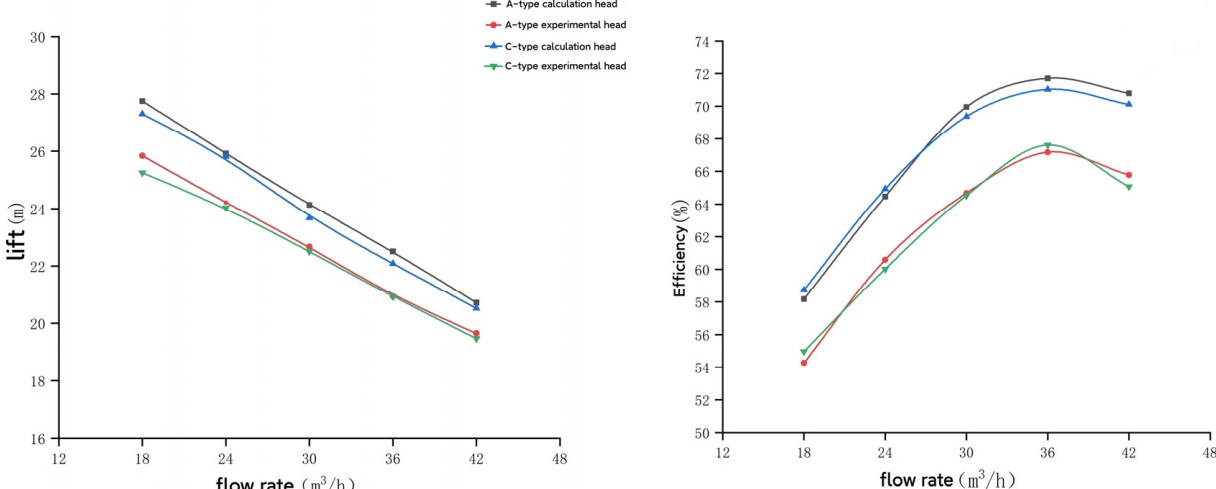

**Figure 17.** Comparison diagram of external characteristic curves of single and double tongued volute centrifugal pumps.

Figure 17 shows that the numerical calculation data of the plastic centrifugal pump with A-type and C-type volute structures are basically consistent with the experimental data. With the increase of the flow rate of the plastic centrifugal pump, the head value of the pump gradually decreases, while the efficiency value of the pump gradually increases and then decreases.

## 5. Double-Tongue Plastic Centrifugal Pump Non-Constant Calculation Analysis

To investigate the influence of the spacing between the worm casing base circle and the tongue of plastic centrifugal pumps on the performance of plastic centrifugal pumps, the unsteady calculation of the pump model of each volute structure is carried out by FLUENT software, and the difference of pressure pulsation between single-tongue volute centrifugal pump and double-tongue volute centrifugal pump under different working conditions is analyzed and compared, and the differences between centrifugal pumps with varying values of the spacing between the worm casing base circle and the tongue [17].

### 5.1. Non-Constant Numerical Calculation Settings

Sliding mesh deals with the coupling between the rotating impeller and stationary volute. The rotational speed of the plastic centrifugal pump is 1450 r/min, that is, 8700 °/s, and the time 0.0003448 s used to rotate the impeller by 3° is selected as a time step. The unsteady numerical calculation simulates the impeller rotation within five weeks after the stable operation of the plastic centrifugal pump and studies and analyzes the results of the last circle, that is, the 481st to 600th time step, the corresponding time is 0.1658488 s to 0.2068800 s [18]. The maximum number of iterations per time step is limited to 30 steps. The SIMPLE method is still selected, and the other parameters remain unchanged. The residual variation process under the design condition is shown in Figure 18.

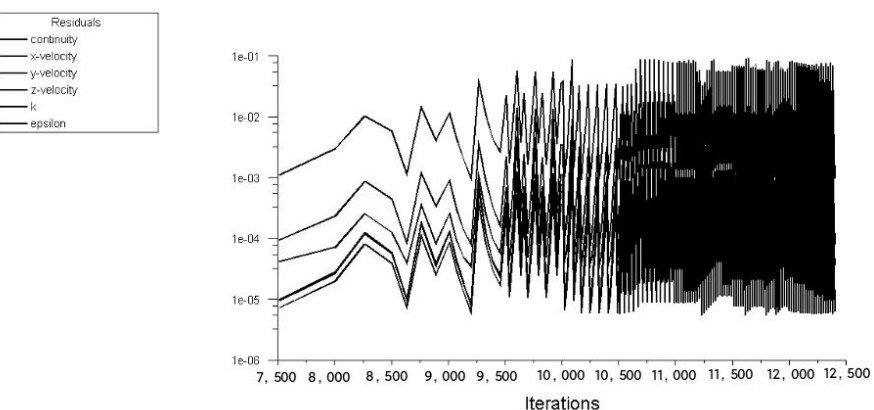

**Figure 18.** Residual monitoring chart.

Pressure pulsation monitoring point setting: four pressure pulsation monitoring points, $P_1$, $P_2$, $P_3$, and $P_4$, are set near the tongue of the inner section of the plastic centrifugal pump, as shown in Figure 19, to monitor the pressure values of each point.

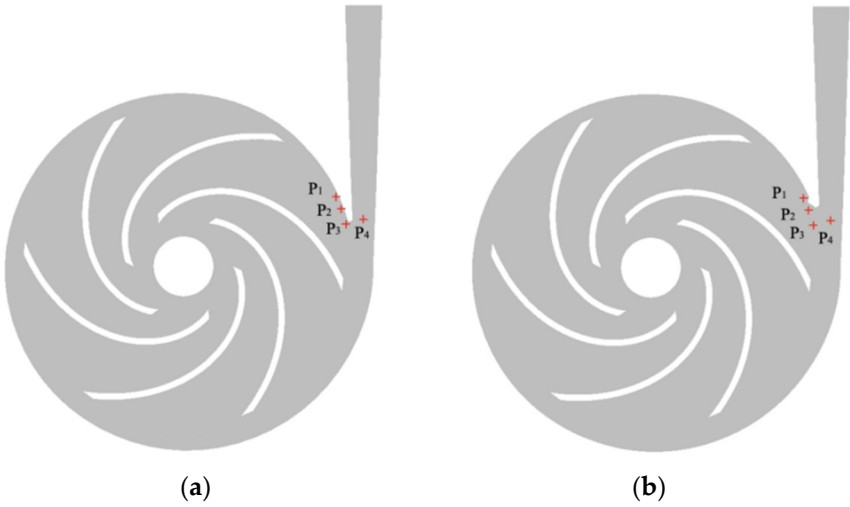

| (**a**) | (**b**) |

**Figure 19.** Pressure pulsation monitoring point location. (**a**) Single septum snail monitoring point location. (**b**) Double septum snail monitoring point location.

*5.2. Analysis of Pressure Pulsation Characteristics under Different Flow Conditions*

1. The 0.8 Q working condition pressure pulsation characteristics analysis

The time and frequency domain plots of pressure pulsations at $P_1$, $P_2$, $P_3$, and $P_4$ for 0.8 Q operating conditions are shown in Figures 20 and 21.

Figure 20 shows that the pressure pulsation at $P_1$ and $P_2$ is more intense at 0.8 Q working conditions, and the pulsation periodicity does not show obviously.

The pressure fluctuation intensity of the single-tongue volute at $P_1$ is higher than that of the double-tongue volute at $P_1$. The pressure fluctuation intensity of the D and E volutes is improved.

The pressure fluctuation intensity of the single-tongue volute at $P_2$ is lower than that of the double-tongue volute at $P_2$. The time domain characteristics of some volutes gradually show periodic variation.

The pressure fluctuation intensity of the single-tongue volute at $P_3$ is lower than that of the double-tongue volute at $P_3$. The time domain characteristics of each volute at $P_3$ gradually show periodic variation.

The pressure fluctuation intensity of the single-tongue volute at $P_4$ is not much different from that of the double-tongue volute at $P_4$. The time domain characteristics of each volute at $P_4$ show significant periodic variation.

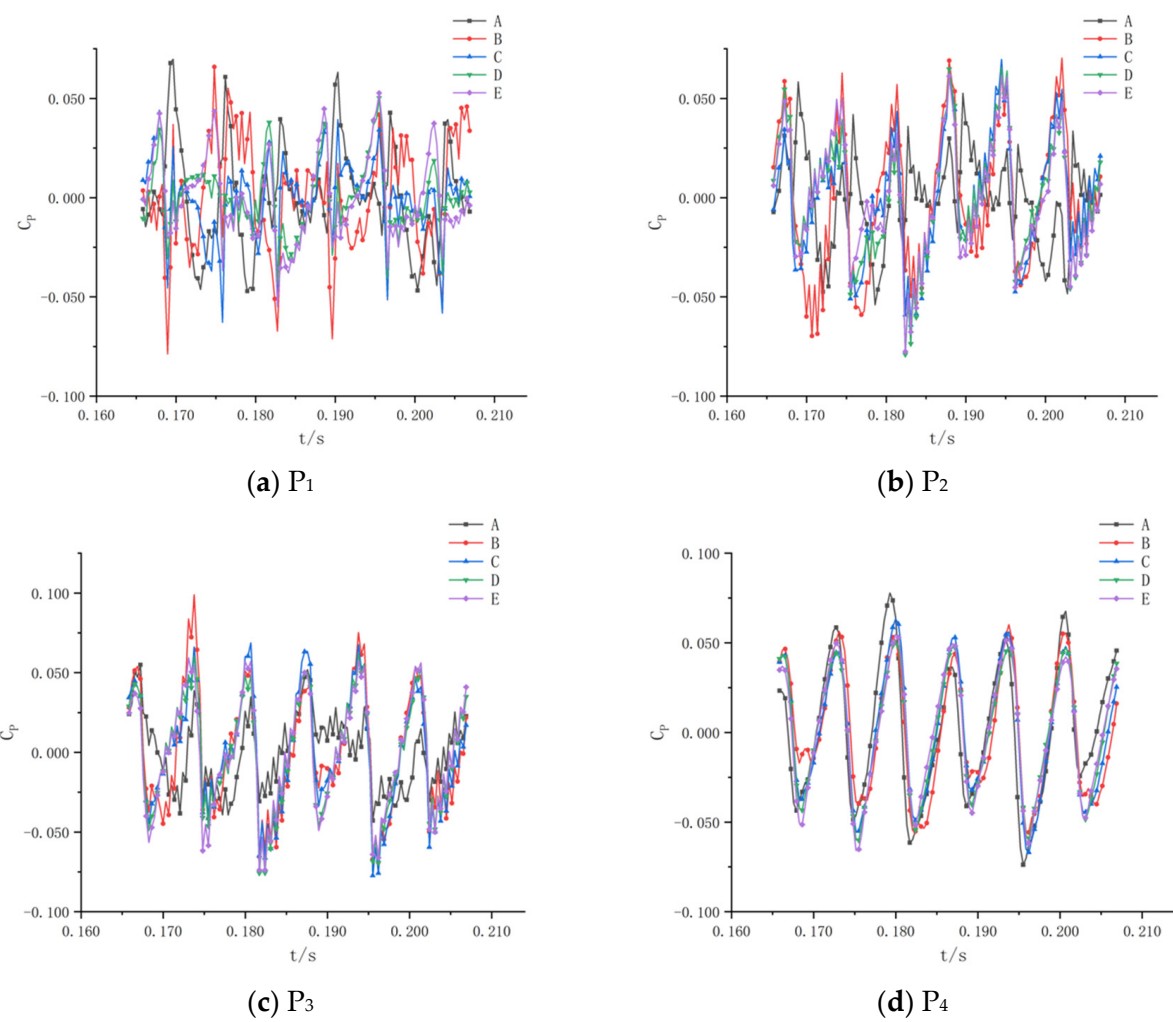

**(a)** $P_1$

**(b)** $P_2$

**(c)** $P_3$

**(d)** $P_4$

**Figure 20.** Time domain diagram of pressure pulsation at 0.8 Q operating conditions.

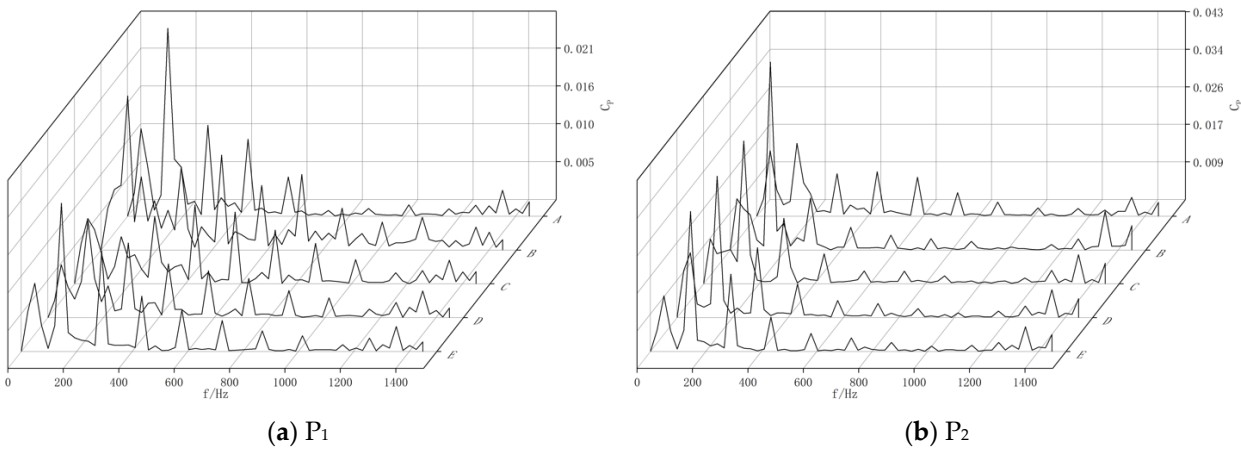

**(a)** $P_1$

**(b)** $P_2$

**Figure 21.** *Cont.*

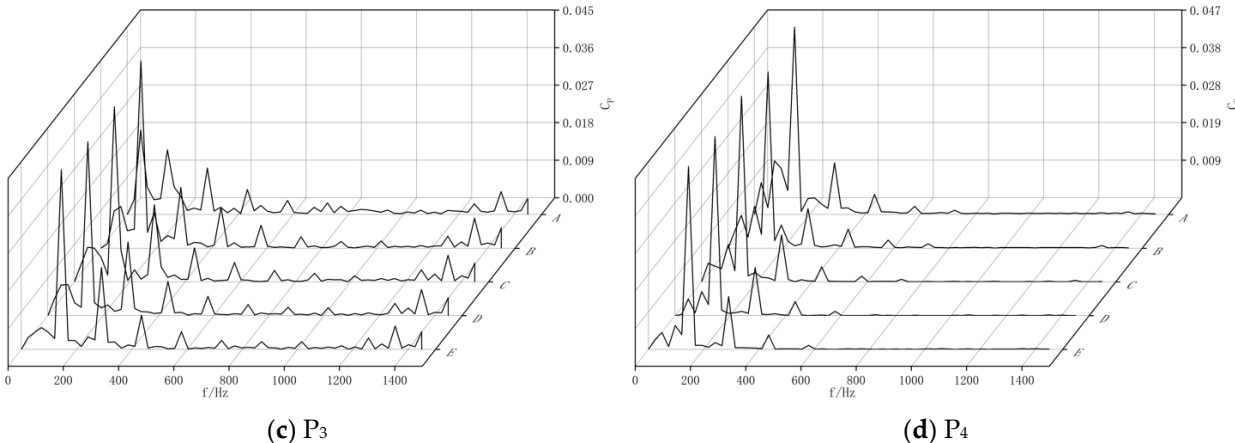

**(c)** $P_3$          **(d)** $P_4$

**Figure 21.** Frequency domain diagram of pressure pulsation at 0.8 Q operating conditions.

Figure 21 and Table 9 show that the central pressure pulsation frequency at four monitoring points, $P_1$, $P_2$, $P_3$, and $P_4$, is similar to the blade passage frequency.

**Table 9.** The primary frequency amplitude of pressure pulsation at each monitoring point under 0.8 Q working conditions.

| Monitoring Points | A Type of Worm Shell | B-Type Worm Shell | C-Type Worm Shell | D-Type Worm Shell | E-Type Worm Shell |
|---|---|---|---|---|---|
| $P_1$ | 0.02588 | 0.01008 | 0.00240 | 0.01332 | 0.02038 |
| $P_2$ | 0.01653 | 0.04263 | 0.03240 | 0.03205 | 0.03174 |
| $P_3$ | 0.01550 | 0.04503 | 0.04200 | 0.04174 | 0.04309 |
| $P_4$ | 0.04701 | 0.04414 | 0.04654 | 0.04487 | 0.04577 |

At $P_1$, the primary frequency amplitude of pressure pulsation of the double-tongue worm shell centrifugal pump is significantly reduced compared to that of the single-tongue worm shell centrifugal pump, with 11%, 91%, 49%, and 21% reduction for each of the B, C, D, and E worm shell centrifugal pumps, respectively.

At $P_2$, the primary frequency amplitude of the pressure pulsation of the double-tongue volute centrifugal pump increased significantly compared to that of the single-tongue volute centrifugal pump, with 158%, 96%, 94%, and 2% increases for each of the B, C, D, and E volute centrifugal pumps, respectively.

At $P_3$, the primary frequency amplitude of the pressure pulsation of the double-tongue volute centrifugal pump increased significantly compared to that of the single-tongue volute centrifugal pump, with the B, C, D, and E volute centrifugal pumps rising by 191%, 171%, 169%, and 178%, respectively.

Since $P_4$ is set farther from the tongue than the other monitoring points, the effect of the change in tongue type on the primary frequency amplitude of the pressure pulsation is not as significant as at the different monitoring points, with the B, C, D, and E volute centrifugal pumps each having a 6%, 1%, 5%, and 3% reduction at $P_4$, respectively.

2. The 1.0 Q pressure pulsation characteristics analysis

The time and frequency domain plots of pressure pulsations at $P_1$, $P_2$, $P_3$, and $P_4$ for 1.0 Q operating conditions are shown in Figures 22 and 23.

Figure 22 shows that the pressure pulsations at $P_1$, $P_2$, and $P_3$ are more intense at 1.0 Q operating conditions. The pressure pulsations at $P_1$, $P_2$, $P_3$, and $P_4$ for single-tongue worm shells and double-tongueworm shells at $P_1$, $P_2$, $P_3$, and $P_4$ do not differ significantly in intensity. The time domain characteristics of each worm shell at $P_1$ and $P_2$ gradually show a periodic variation pattern, and those at $P_3$ and $P_4$ already show a significant irregular variation pattern.

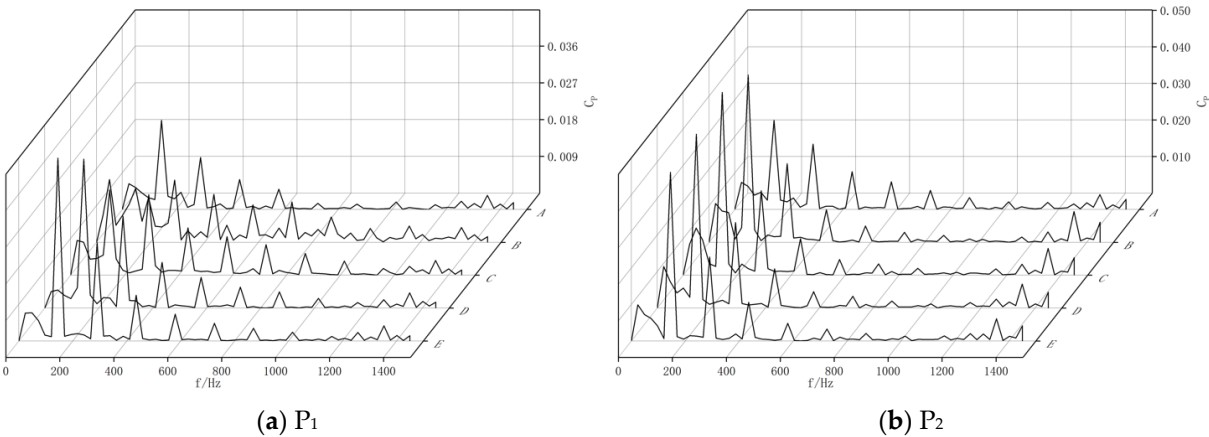

**(a)** P$_1$      **(b)** P$_2$

**(c)** P$_3$      **(d)** P$_4$

**Figure 22.** Time domain diagram of pressure pulsation at 1.0 Q operating conditions.

**(a)** P$_1$      **(b)** P$_2$

**Figure 23.** *Cont.*

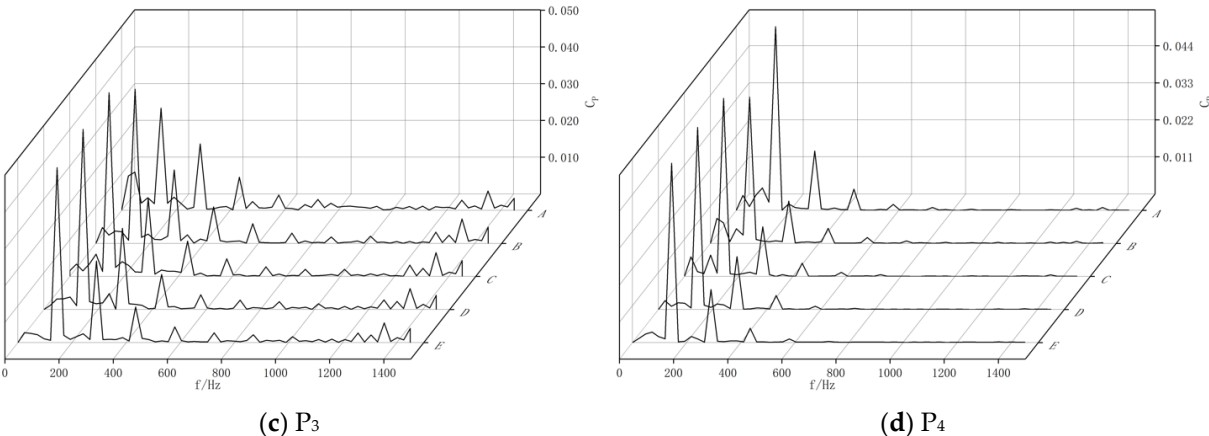

**(c)** $P_3$                                              **(d)** $P_4$

**Figure 23.** Frequency domain diagram of pressure pulsation at 1.0 Q operating conditions.

Figure 23 and Table 10 show that the central frequency of pressure pulsation at four monitoring points, $P_1$, $P_2$, $P_3$, and $P_4$, is related to the blades' pressure pulsation through similar frequencies.

**Table 10.** Main frequency amplitude of pressure pulsation at each monitoring point under 1.0 Q working conditions.

| Monitoring Points | A-Type Worm Shell | B-Type Worm Shell | C-Type Worm Shell | D-Type Worm Shell | E-Type Worm Shell |
|---|---|---|---|---|---|
| $P_1$ | 0.02156 | 0.01308 | 0.02064 | 0.03606 | 0.04433 |
| $P_2$ | 0.02439 | 0.04579 | 0.05011 | 0.04756 | 0.04615 |
| $P_3$ | 0.02786 | 0.04197 | 0.05011 | 0.04906 | 0.04706 |
| $P_4$ | 0.05471 | 0.04346 | 0.05300 | 0.05413 | 0.05330 |

At $P_1$, the primary frequency amplitude of the pressure pulsation of the B and C worm shell centrifugal pumps decreased significantly compared to that of the single-tongue worm shell centrifugal pump by 39% and 4%, respectively, while the primary frequency amplitude of the pressure pulsation of the D and E worm shell centrifugal pumps increased significantly compared to that of the single-tongue worm shell, by 67% and 106%, respectively.

At $P_2$, the primary frequency amplitude of pressure pulsation of the double-tongue worm shell centrifugal pump increased significantly compared to that of the single-tongue worm shell centrifugal pump, with 88%, 105%, 95%, and 89% increases for each of the B, C, D, and E worm shell centrifugal pumps, respectively.

At $P_3$, the primary frequency amplitude of the pressure pulsation of the double-tongue volute centrifugal pump increased significantly compared to that of the single-tongue volute centrifugal pump, with 51%, 80%, 76%, and 71% increases for each of the B, C, D, and E volute centrifugal pumps, respectively.

Since $P_4$ is set farther from the tongue than the other monitoring points, the effect of the change in tongue type on the primary frequency amplitude of the pressure pulsation is not as significant as at the different monitoring points, with 21%, 3%, 1%, and 3% reductions at $P_4$ for each of the B, C, D, and E volute centrifugal pumps, respectively.

3. The 1.2 Q pressure pulsation characteristics analysis

The time and frequency domain plots of pressure pulsations at $P_1$, $P_2$, $P_3$, and $P_4$ for 1.2 Q operating conditions are shown in Figures 24 and 25.

Figure 24 shows that the pressure pulsations at $P_1$, $P_2$, and $P_3$ are more intense at 1.2 Q operating conditions. The pressure pulsations at $P_1$, $P_2$, $P_3$, and $P_4$ for single-tongue worm shells are similar to those at $P_1$, $P_2$, $P_3$, and $P_4$ for double-tongue worm shells. The time

domain characteristics of each worm shell at $P_1$, $P_2$, $P_3$, and $P_4$ show a significant periodic variation pattern.

**(a)** $P_1$

**(b)** $P_2$

**(c)** $P_3$

**(d)** $P_4$

**Figure 24.** Time domain diagram of pressure pulsation at 1.2 Q operating conditions.

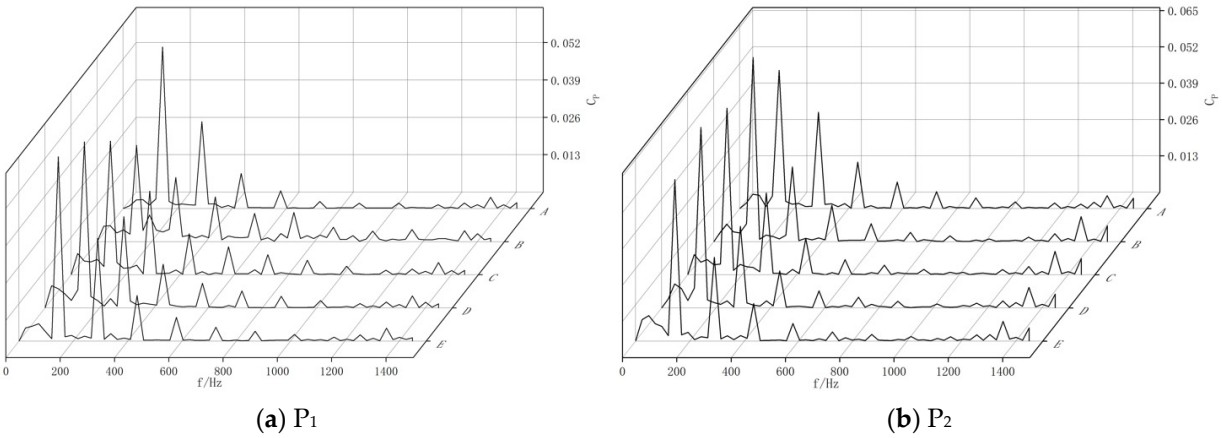

**(a)** $P_1$

**(b)** $P_2$

**Figure 25.** *Cont.*

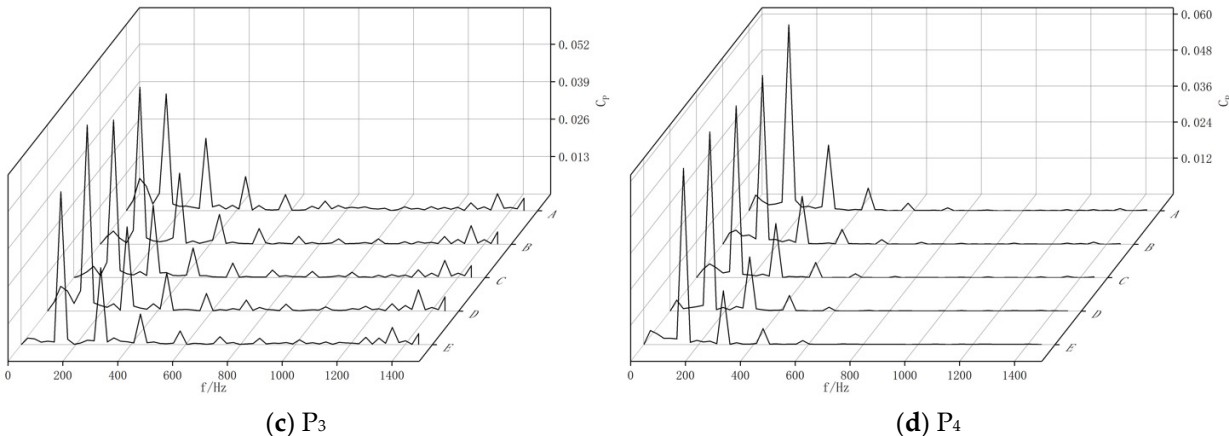

**(c)** $P_3$                        **(d)** $P_4$

**Figure 25.** Frequency domain diagram of pressure pulsation at 1.2 Q operating conditions.

Figure 25 and Table 11 show that the central pressure pulsation frequency at the four monitoring points, $P_1$, $P_2$, $P_3$, and $P_4$, is similar to the blade passage frequency.

**Table 11.** The 1.2 Q working conditions at each monitoring point pressure pulsation main frequency amplitude table.

| Monitoring Points | A-Type Worm Shell | B-Type Worm Shell | C-Type Worm Shell | D-Type Worm Shell | E-Type Worm Shell |
|---|---|---|---|---|---|
| $P_1$ | 0.05627 | 0.03355 | 0.04656 | 0.05782 | 0.06422 |
| $P_2$ | 0.04948 | 0.06610 | 0.05970 | 0.06468 | 0.05785 |
| $P_3$ | 0.04054 | 0.05444 | 0.05474 | 0.06468 | 0.05302 |
| $P_4$ | 0.06215 | 0.05625 | 0.05730 | 0.05978 | 0.05880 |

At $P_1$, the primary frequency amplitude of the pressure pulsation of the B and C volute centrifugal pumps decreased significantly compared to that of the single-tongue volute centrifugal pump by 40% and 17%, respectively, while the primary frequency amplitude of the pressure pulsation of the D and E volute centrifugal pumps increased significantly compared to that of the single-tongue volute centrifugal pump, by 3% and 14%, respectively.

At $P_2$, the primary frequency amplitude of pressure pulsation of the double-tongue volute centrifugal pump increased significantly compared to that of the single-tongue volute centrifugal pump, with 34%, 21%, 31%, and 17% increases for each of the B, C, D, and E volute centrifugal pumps, respectively.

At $P_3$, the primary frequency amplitude of the pressure pulsation of the double-tongue volute centrifugal pump increased significantly compared to that of the single-tongue volute centrifugal pump, with 34%, 35%, 60%, and 31% increases for each of the B, C, D, and E volute centrifugal pumps.

Since $P_4$ is set farther from the tongue than the other monitoring points, the effect of the change in tongue type on the primary frequency amplitude of the pressure pulsation is not as significant as at the different monitoring points, with the B, C, D, and E volute centrifugal pumps decreasing by 9%, 8%, 4%, and 5% each at $P_4$, respectively.

## 6. Conclusions

In this paper, five kinds of volute structures are designed. FLUENT is used to calculate the steady and unsteady calculation of the five volute structure models. The static pressure, velocity, turbulent kinetic energy, and impeller radial force under different working conditions are used as performance indicators. It is found that the static pressure gradient distribution of the double-tongue volute centrifugal pump under various working conditions is improved to varying degrees, and the low-speed region of the diffusion section is reduced. The double-tongue volute centrifugal pump can reduce the maximum turbulent

kinetic energy value at the tongue to a certain extent under small flow conditions and can reduce the impeller radial force value to a certain extent and balance part of the radial force. Taking the pressure pulsation intensity as the performance index, it is found that with the increase of flow rate, the periodic variation of the time domain characteristics of the pressure pulsation intensity at the monitoring point is gradually obvious. Under the working conditions of 0.6 Q, 0.8 Q, and 1.2 Q, the pressure pulsation intensity at the P1 and P2 monitoring points of the double-tongue volute centrifugal pump is lower than that of the single-tongue volute centrifugal pump, and the amplitude of the main frequency of the pressure pulsation at the P1, P2, and P4 monitoring points is reduced. The influence of double-tongue volute structure on the performance of plastic centrifugal pump was studied in this research.

**Author Contributions:** Z.R. offered substantial contributions to design, experimental research, data collection, and result analysis; L.T. made critical changes to important academic content; H.Z. conducted the final review and finalization of the articles to be published. All authors have read and agreed to the published version of the manuscript.

**Funding:** This article belongs to the project of the "The University Synergy Innovation Program of Anhui Province (GXXT-2019-004)" "Natural Science Research Project of Anhui Universities (KJ2021ZD0144)" "Science and Technology Planning Project of Wuhu City (2021YF58)".

**Institutional Review Board Statement:** Not applicable.

**Informed Consent Statement:** Not applicable.

**Data Availability Statement:** The data used to support the findings of this study are included within the article.

**Conflicts of Interest:** The authors declare that they have no conflict of interest to report regarding the present study.

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
