# Peer review of "Double-Tongue Worm Shell Structure on Plastic Centrifugal Pump Performance Study"

_applsci, doi:10.3390/app13148507_

Round 1

Reviewer 1 Report

The paper deals with numerical analysis of plastic cenrifugal pump performance study. Results of extremally detailed analysis are presented in the paper.

The amount of work done is admirable, however the paper is not ready for publication, for two main reasons:

1. Experimental validation is not provided. For this reason it is not possible to fully trust the results obtained

2. I suggest to divide the paper and submited the paper as part I and part II. The amount of information contained in the manuscript makes this paper difficult to digest

3. Abstract should be modified. Authors are asked to provide in short manner the description of the problem, the method used and obtained results. In the abstract some quantitative parameters should be used. By this way the reader will be informed about the content of the paper. 

Reviewer 2 Report

 “UG, ICEM, FLUENT” must be declared in the beginning. 

I assume Table 1 will not be split in the final version

The aspect ratio of Figure 2 is wrong.

It’s very hard to realize the difference between Type A – E in Figure 4, and in Figures 8-16. The authors must clarify the slight variances in the text. 

Figure 7: The x-title “Number of grids” sounds bad. I suggest “Number of cells”

 “UG, ICEM, FLUENT” must be declared in the beginning. 

I assume Table 1 will not be split in the final version

The aspect ratio of Figure 2 is wrong.

It’s very hard to realize the difference between Type A – E in Figure 4, and in Figures 8-16. The authors must clarify the slight variances in the text. 

Figure 7: The x-title “Number of grids” sounds bad. I suggest “Number of cells”

Reviewer 3 Report

Comments to Authors:

The manuscript as provided is not up to the mark of the journal and requires improvements.

This work is involved the following issues which need to be resolved very carefully. 

1. Abstract should be rewritten highlighting the novel quantitative findings.

2. The authors need to highlight the specific applications of this study in real-world

engineering and industry. Authors should cite at least one concrete potential application

where the present results are likely to be relevant, howsoever remotely, over this range of

conditions including the boundary conditions adopted.

3. The novelty of the work needs to be more explicitly stated through extensive literature

reviews. There are significant volumes of works on the above subject topic flow in

different geometries in the published literature.

4. The model validation study should be rigorous (!) and detailed (!) with experiments, at

least for a related problem. This is non-negotiable. 

5. The results and discussion section is suffering from in-depth flow physics and this section

is written very poorly. In your discussion section, please link your empirical results with a

broader and deeper literature review.

6. The conclusion should be re-written highlighting the novel findings alone.

7. There are significant grammatical and language errors/ mistakes throughout the

manuscript.

8. List out all the symbols in the nomenclature section.

Round 2

Reviewer 1 Report

The manuscript was revised. The reviewer suggestions have been taken into consideration. Authors have provided answers for Reviewer doubts.

Please complete the following gaps:

- In fig.2. the title of axis is missing

- plot legend on Fig 17 is required

Reviewer 3 Report

accept in current form

Author Response

Thank you very much for your comments, and thank you for your approval of our article.